# An Analysis of Virtual Nodes in Graph Neural Networks for Link Prediction

**EunJeong Hwang**
University of British Columbia
ejhwang@cs.ubc.ca

**Veronika Thost**[*]
MIT-IBM Watson AI Lab
Veronika.Thost@ibm.com

**Shib Sankar Dasgupta**
University of Massachusetts Amherst
ssdasgupta@umass.edu

**Tengfei Ma**[*]
IBM Research
Tengfei.Ma1@ibm.com

## 1 Introduction

It is well known that the graph classification performance of graph neural networks (GNNs) often improves by adding an artificial *virtual node* to the graphs, which is connected to all graph nodes [1–4]. While virtual nodes were originally thought as aggregated representations of the entire graph, they also provide shortcuts for message passing between nodes along the graph edges. Surprisingly, *the advantages of virtual nodes have never been theoretically investigated, and their impact on other problems is still an open research question.* We adapt and study the virtual node concept for problems over networks, which are usually larger, often very sparse or dense, and overall more heterogeneous.

Many popular GNNs are based on message passing, which computes node embeddings by iteratively aggregating the features of (usually direct) neighbor nodes along the graph edges [1]. In this way, they are able to distinguish (non-)isomorphic nodes (to great extent) [5], but this does not transfer to links [6]; for links, extra procedures may be needed (e.g., modeling enclosing subgraphs [7]). Furthermore, *on large graphs, GNNs may face the under-reaching problem* if long-range dependencies beyond the model's computing radius are important for the problem at hand (e.g., complex chains of protein-protein interactions). *Over dense graphs, GNNs with many layers struggle with over-smoothing*, node representations converging to similar values. There have been several proposals to overcome these problems. On the one hand, several works propose techniques that allow for larger numbers of GNN layers [8–14]. However, as shown in our later results, many of them do not perform well on link predictions tasks, especially on comparably dense graphs. On the other hand, there are approaches that adapt message passing to consider neighbors beyond the one-hop neighborhood: based on graph diffusion [15–20] and other theories [21, 22]. Yet, most of these models are relatively complex and, in fact, in our experiments over the challenging graphs from the Open Graph Benchmark (OGB) [23], several ran out of memory. In this paper, *we show that virtual nodes may alleviate these typical issues of GNNs over larger graphs.*

We focus on link prediction (but most results may be easily extended to node classification), which is important in view of incomplete graph data in practice in various different domains [24–27] . Numerous models have been proposed to solve this problem in the past, ranging from knowledge-graph-specific predictors [27] to GNNs [7, 24]. We explore the application and effects of virtual nodes in link prediction both theoretically and empirically:

- *We propose to use multiple virtual nodes in the network graph scenario and describe a graph-based technique to connect them to the graph nodes.* In a nutshell, we use a graph clustering algorithm to determine groups of nodes in the graph belonging together and then connect these nodes to a common virtual node (see Figure 1). In this way, we add expressiveness, and under-reaching is decreased because clustered nodes can share information easily; meanwhile, the nodes are spared of unnecessary information from unrelated nodes (i.e., in contrast to the single virtual node model).

---

[*]Correspondence to Tengfei Ma, Veronika Thost

Hwang et al., An Analysis of Virtual Nodes in Graph Neural Networks for Link Prediction (Extended Abstract). Presented at the First Learning on Graphs Conference (LoG 2022), Virtual Event, December 9–12, 2022.

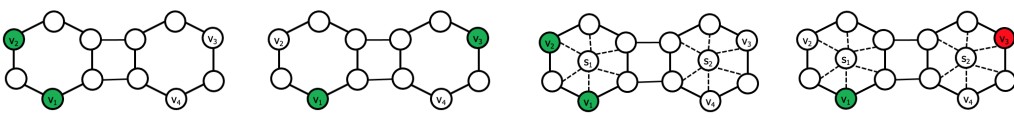

(a) Without virtual nodes.  (b) With two virtual nodes.

**Figure 1:** Multiple virtual nodes increase expressiveness: a regular GNN computes the same representation for isomorphic nodes $v_2$ and $v_3$, and hence cannot discriminate links $\{v_1, v_2\}$ and $\{v_1, v_3\}$. The embeddings of the nodes can be influenced by virtual nodes $s_1$ and $s_2$ and become different.

- We also investigate alternative methods to determine the virtual node connections (i.e., randomization) and compare to the original model with a single virtual node.
- We provide the first *theoretical analysis of the benefits of virtual nodes* in terms of (I) expressiveness of the learned link representation and (II) potential impact on under-reaching and over-smoothing.
- We conducted experiments over challenging datasets, provide ablation studies and a detailed analysis of important factors.

Altogether, we show that, also for link prediction, *virtual nodes are simple but powerful extensions that may yield rather stable performance increases* for various standard GNNs. Since the latter represent simple and proven models which are especially interesting for applications, our study provides practical guidance and explanations about where and how virtual nodes may provide benefits. In this abstract, we give an overview of our main findings; for details and additional results, see the appendix. Our code is available at `https://github.com/eujhwang/vn-analysis`.

## 2   Preliminaries

**Link Prediction.** We consider a *graph* $G = (V, E)$ with *nodes* $V$ and undirected *edges* $E \subseteq V \times V$. This basic choice is only for ease of presentation; our techniques work for directed graphs and (with simple adaptation) for graphs with labelled nodes (edges). We assume $V$ to be ordered and may refer to a node by its index in $V$. For a node $v \in V$, $\mathcal{N}_v$ denotes the set of its neighbors. Given two nodes, the *link prediction task* is to predict if there is a link between them.

**Message-Passing Graph Neural Networks.** In this paper, we usually use the term *graph neural networks* (GNNs) to denote GNNs that use message passing as described in [1]. These networks compute for every $v \in V$ a node representation $h_v^\ell$ at layer $\ell$, by *aggregating* its neighbor nodes based on a generic aggregation function and then *combine* the obtained vector with $h_v^{\ell-1}$ as below.

$$h_v^\ell = \text{COMB}^\ell\Big(h_v^{\ell-1}, \text{AGG}^\ell\big(\{h_u^{\ell-1} \mid u \in \mathcal{N}_v\}\big)\Big) \tag{1}$$

Link prediction with GNNs is usually done by combining the final representations of nodes $u, v$ under consideration and passing them through several feed-forward layers with a final sigmoid function for scoring. Our implementation follows this approach.

## 3   Virtual Nodes in Graph Neural Networks for Link Prediction

**Multiple Virtual Nodes.** The intuition of using virtual nodes is to provide a shortcut for sharing information between the graph nodes. However, the amount of information in a graph with possibly millions of nodes is enormous, and likely too much to be captured in a single virtual node embedding. Further, not all information is equally relevant to all nodes. Therefore we suggest to use *multiple virtual nodes* $S = \{s_1, s_2 \ldots, s_n\}$[2] each being connected to a subset of graph nodes, as determined by an assignment $\sigma : V \to [1, n]$; $n$ is a hyperparameter. We consider two options to obtain $\sigma$:

**Randomness: GNN-RM.** Most simply, we can determine a fixed $\sigma$ randomly once with initialization.

**Clustering: GNN-CM.** Many types of graph data incorporate some cluster structure that reflects which nodes belong closely together (e.g., collaboration or social networks). We propose to connect

---

[2]Since notation $V$ is standard for nodes, we use $S$ for the set of virtual nodes. Think of "supernodes".

nodes in such a cluster to a common virtual node, such that the structure inherent to the given graph is reflected in our virtual node assignment $\sigma$. More precisely, during initialization, we use a generic clustering algorithm (e.g., METIS [28]) which, given a number $m$, creates a set $C = \{C_1, C_2 \ldots, C_m\}$ of clusters (i.e., sets of graph nodes) by computing an assignment $\rho : V \to [1, m]$, assigning each graph node to a cluster. We then set $m = n$ and $\sigma = \rho$.

**The Model.** We integrate the multiple virtual nodes into a generic message-passing GNN in a straightforward way extending the approach from [23] to include multiple virtual nodes, computing node representations $h_v^\ell$ for a node $v \in V$ at layer $\ell$ as below. The highlighted adaptation of the standard GNN (Equation (1)) is only minor, but powerful. In our implementation, $\text{COMB}_{\text{VN}}^\ell$ is addition combined with linear layers and layer normalization, $\text{AGG}_{\text{VN}}^\ell$ is a sum.

$$h_{s_i}^\ell = \text{COMB}_{\text{VN}}^\ell \Big( h_{s_i}^{\ell-1}, \text{AGG}_{\text{VN}}^\ell \big( \{ h_u^{\ell-1} \mid u \in V, \sigma(u) = i \} \big) \Big)$$

$$h_v^\ell = \text{COMB}^\ell \Big( h_v^{\ell-1} + h_{s_{\sigma(v)}}^\ell, \text{AGG}^\ell \big( \{ h_u^{\ell-1} \mid u \in \mathcal{N}_v \} \big) \Big)$$

### 3.1 Analysis I: Virtual Nodes Increase Expressiveness

Additional structure-related features such as distance encodings [6, 29] are known to make graph representation learning more powerful. Our multiple virtual notes have a similar effect. Figure 1 gives an intuition of how they can increase the expressiveness of the regular 1-WL-GNN. Consider the nodes $v_2, v_3$ with the same local structure, which means they can get the same node representations by using 1-WL-GNN. So we cannot discriminate the links $(v_1, v_2)$ and $(v_1, v_3)$ if we just use 1-WL-GNN and concatenate the node representations for link prediction. However, if we add 2 virtual nodes and add extra features to each node. $v_1$ and $v_2$ get a new feature $(1, 0)$, $v_3$ get new feature $(0, 1)$. So it is easy to see $(v_1, v_2)$ and $(v_1, v_3)$ now have different representations.

Furthermore, we have the following theorem to show that using multiple virtual nodes can increase the power of discriminating links. More details are in Appendix B.1.

**Theorem 3.1** *Given an arbitrary non-attributed graph with $n$ nodes, a constant $\epsilon > 0$, and $m$ virtual nodes that evenly divide the node set into $m$ clusters, we have: If the degree of each node in the graph is between $1$ and $\mathcal{O}(\log^{\frac{1-\epsilon}{2k}}(n))$, then there are $\omega\left((m-1)^2 (\frac{n^\epsilon}{m}-1)^3\right)$ many pairs of non-isomorphic links $(u, w), (v, w)$ such that a $k$-layer 1-WL-GNN gives $u, v$ the same representation, while using $m$ virtual nodes give $(u, w), (v, w)$ different representations.*

### 3.2 Analysis II: Virtual Nodes Impact Node Influence

We assume we can learn useful embeddings for virtual nodes if the assignment is chosen appropriately. Based on the above analysis, we expect virtual nodes to positively impact learning and prediction performance. Following [8, 16], we measure the sensitivity of a node $y$ on a node $x$ by the *influence score*. For a $k$-layer GCN, this score is known to be proportional in expectation to the $k$-*step random walk distribution* $P_{rw}$ from $x$ to $y$.[3] We exploit this relationship and argument in terms of $P_{rw}$.

**Impact of Virtual Nodes.** For simplicity, we consider the influence score in an $r$-regular graph. Consider the message passing between two nodes $x$ and $y$. For $k = 1$, all the nodes can be classified into two cases: if $y$ is not connected to $x$, the influence changes from $0$ to $\frac{1}{(r+2)}$; otherwise:

$$P_{rw}^s(x \to y, 1) = P_{rw}(x \to y, k) + P_{rw}(x \to s, s \to y) = \frac{1}{(r+2)} + \frac{1}{(r+2)|V|}.$$

For $k >= 2$, by adding a virtual node $s$ in one GNN layer, the probability changes to:

$$P_{rw}^s(x \to y, k) = P_{rw}(x \to y, k) + P_{rw}(x \to s, s \to y) P_{rw}(x \to y, k-1)$$

$$= \frac{|R^k|}{(r+2)^k} + \frac{1}{(r+2)|V|} P_{rw}^s(x \to y, k-1).$$

---

[3]See Theorem 1 in [8]; that theorem makes some simplifying assumptions (e.g., on the shape of GCN).

**Table 1:** Comparison of virtual-node augmented GNNs to models with similar goal; $*$: from OGB leaderboard.

| | ddi Hits@20 | collab Hits@50 |
|---|---|---|
| SEAL$^*$ | $30.56 \pm 3.86$ | $\mathbf{64.74 \pm 00.43}$ |
| DeeperGCN$^*$ | n/a | $52.73 \pm 00.47$ |
| SGC | $06.76 \pm 05.86$ | $46.35 \pm 01.97$ |
| P-GNN | $10.50 \pm 00.00$ | mem. |
| APPNP | $14.92 \pm 02.98$ | $31.85 \pm 02.05$ |
| GCN | $40.76 \pm 10.73$ | $49.55 \pm 00.64$ |
| GCN-GDC | $25.50 \pm 12.42$ | mem. |
| GCN+JKNet$^*$ | $60.56 \pm 08.69$ | n/a |
| GCN+LRGA$^*$ | $62.30 \pm 09.12$ | $52.21 \pm 00.72$ |
| GCN-VN | $62.17 \pm 12.41$ | $50.49 \pm 00.88$ |
| GCN-RM | $55.32 \pm 12.62$ | $50.83 \pm 01.09$ |
| GCN-CM | $61.05 \pm 15.63$ | $51.81 \pm 00.76$ |
| SAGE | $61.73 \pm 10.68$ | $55.16 \pm 01.71$ |
| SAGE-GDC | $31.41 \pm 12.54$ | mem. |
| SAGE+edges$^*$ | $74.95 \pm 03.17$ | n/a |
| SAGE-VN | $64.91 \pm 13.60$ | $58.75 \pm 00.91$ |
| SAGE-RM | $\underline{70.68 \pm 11.74}$ | $58.30 \pm 00.87$ |
| SAGE-CM | $\mathbf{76.21 \pm 11.57}$ | $\underline{60.17 \pm 01.37}$ |

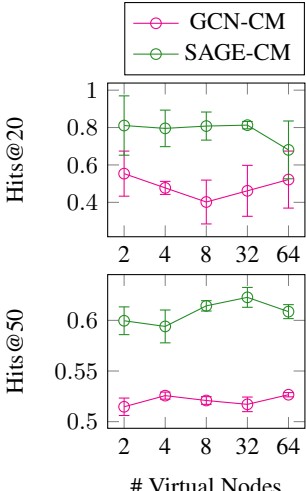

**Figure 2:** Impact of virtual node number; ddi (top) and collab (bottom).

**Multiple Virtual Nodes.** We continue along these lines and assume there is a shortest path of length $\leq k$ between $x$ and $y$. If $x$ and $y$ connect to the same virtual node $s$, then the above changes to:

$$P_{rw}^s(x \to y, k) = \frac{|R^k|}{(r+2)^k} + \frac{1}{(r+2)|C_s|} P_{rw}^s(x \to y, k-1). \tag{2}$$

Since the set $C_s$ of nodes connecting to $s$ is much smaller than $V$, *multiple virtual nodes can increase the impact of potentially important distant nodes more than a single virtual node*.

**Impact on Over-Smoothing** The idea is to show that multiple virtual nodes help preserve more local information. If we consider the influence of $x$ onto itself, we can show that, *with a single virtual node, a graph node can preserve less information for itself* at each layer. However, this changes in view of multiple virtual nodes; in particular, when $|C_s| \leq r+1$. We encounter this scenario practically especially with dense graphs. This fits nicely since dense graphs are particularly prone to over-smoothing and, as shown in [8, 16], *additional capability to preserve local information in message passing steps helps to reduce over-smoothing*. More details are shown in Appendix B.2.

## 4    Evaluation

**Datasets.** We use two datasets from OGB: ddi, a drug-drug interaction network; and collab, an author collaboration network [23]. Data statistics are in Appendix (Table 4). ddi is dense with a low graph diameter; while collab is sparser with large diameter. Both have high clustering coefficients.

**Models.** Standard GNNs: **GCN** [30] and **SAGE** [31], which we extend with virtual nodes; deep GNNs: **SGC**, **APPNP**, **DeeperGCN**, GCN-**JKNet**; message passing beyond the direct neighborhood: **P-GNN** [22], **APPNP** [16], **GDC** [20]; and an advanced GNN-based link predictor: **SEAL** [7].

**Results, Table 1. Overall Impact of Virtual Nodes.** The common approach of using a single virtual node (GNN-VN) yields good improvements over ddi and slight improvements over collab. The numbers for GNN-RM reflect the randomness of their connections to the virtual nodes, there is no clear trend; but they clearly outperform the original models. *The virtual node assignment based on the graph structure (GNN-CM) yields consistently good improvements over ddi and collab.* We note that we obtained some ambiguous results on other smaller data that has less cluster structure, but overall can observe a positive impact.

**Model Comparison.** The results of the best models from the OGB leaderboard vary strongly with the different datasets (e.g., SEAL), or have not been reported at all. Most deep GNNs and models that use complex message-passing techniques perform disappointing and, overall, much worse than the standard GNNs. We did thorough hyperparameter tuning for these models and it is hard to explain. A possible reason may be most of their original evaluations focus on node or graph classification and consider very different types of data. The model closest to our approach is the position-aware graph neural network (P-GNN) [22]. It assigns nodes to random subsets of nodes called "anchor-sets", and then learns a non-linear aggregation scheme that combines node feature information from each anchor-set and weighs it by the distance between the node and the anchor-set. So, it creates a message for each node for every anchor-set, instead of for each direct neighbor. The fact that it ran out of memory on `collab` shows that practice may benefit from simpler or more efficient schemes.

**Impact of Virtual Node Number, Figure 2.** The configurations of the best models provided in the appendix show that the chosen numbers of virtual nodes are indeed random for the "random" models, but GNN-CM consistently uses a high number of virtual nodes, which also suits it better according to our theoretical analysis. In line with this, the more detailed analysis varying the numbers of virtual nodes, yields best results (also in terms of standard deviations) for SAGE-CM at rather high values. For GCN, we do not see a clear trend, but (second) best performance with 64 virtual nodes.

**Table 2:** Comparison of using the virtual nodes at every and only at the last layer; Hits@20, `ddi`.

|  | GCN | SAGE | GIN |
|---|---|---|---|
| w/o virtual nodes | $0.5062 \pm 0.2186$ | $0.6128 \pm 0.2122$ | $0.4829 \pm 0.1608$ |
| - VN | $0.5932 \pm 0.2390$ | $\underline{0.7160 \pm 0.1457}$ | $\underline{0.6523 \pm 0.0446}$ |
| - $VN_{OL}$ | $0.6180 \pm 0.0088$ | $0.5167 \pm 0.1364$ | $0.6472 \pm 0.0542$ |
| - CM | $\underline{0.6322 \pm 0.1565}$ | $\mathbf{0.8819 \pm 0.0341}$ | $\mathbf{0.6544 \pm 0.0960}$ |
| - $CM_{OL}$ | $\mathbf{0.6338 \pm 0.1188}$ | $0.6151 \pm 0.1545$ | $0.4420 \pm 0.1694$ |

**Using Virtual Nodes Only at the Last GNN Layer, Table 2.** [32] show that using a fully connected adjacency matrix at the last layer of a standard GNN helps to better capture information over long ranges. We therefore investigated if it is a better architectural choice to use virtual nodes only at the last layer. However, we observed that this can lead to extreme performance drops. In Table 2, $VN_{OL}$ and $CM_{OL}$ indicate the ablation models which use virtual nodes only at the last layer. It shows that these ablations models decrease the performance a lot. That means using virtual nodes only at the last layer is not enough.

**Impact of Clustering Algorithm, Table 3.** Our architecture is generic in the clustering algorithm, and we investigated the effects of varying that. Note that $CM_{metis}$ in the table is our GNN-CM model used in Table 1. Graclus is similar in nature to METIS in that it also creates partitions based on the adjacency matrix, but it took much longer to run. Diffpool considers the node features and yields improvements for GCN, but does not scale to larger datasets. Over `ddi`, there is no clear winner and, given its efficiency, METIS turns out to be a good solution.

**Table 3:** Comparison of clustering algorithms to determine virtual node connections; Hits@20, `ddi`. See analysis in the main paper.

|  | GCN | SAGE | GIN |
|---|---|---|---|
| $CM_{metis}$ | $0.6322 \pm 0.1565$ | $\mathbf{0.8819 \pm 0.0341}$ | $0.6544 \pm 0.0960$ |
| $CM_{graclus}$ | $0.6324 \pm 0.1048$ | $0.5406 \pm 0.3769$ | $\mathbf{0.7299 \pm 0.0604}$ |
| $CM_{diffpool}$ | $\mathbf{0.7392 \pm 0.0381}$ | $0.7556 \pm 0.1252$ | $0.5436 \pm 0.1572$ |

**Conclusions and Discussions.** In a nutshell, our clustering-based virtual node assignment provides stable performance increases if the graph contains good cluster structure and is sufficiently large. In smaller graphs, the GNNs alone were usually sufficient. In line with our theoretical investigation, we expect virtual nodes to be especially beneficial over dense graphs.

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

## A  Appendix

## B  Additional Theoretical Results

### B.1  Expressiveness of link representation

Given multiple virtual nodes $S = \{s_1, ..., s_m\}$, we obtain a node labeling that includes the node representations $h^\ell_{s_i}$ of the virtual nodes. For every $u \in V$, we have additional features $l(u|S) = (h^\ell_{s_1}, ..., h^\ell_{s_m})^{\mathrm{T}}(\gamma(u|s_1), ..., \gamma(u|s_m))$, where $\gamma(u|s_i) = 1$ if $u$ is connected to the virtual node $s_i$, and $\gamma(u|v_i) = 0$ otherwise. We can initialize $h^0_{s_i}$ with the one-hot encoding of $i$ to ensure the $s_i$ have different labels. We can then show that this labeling increases the power of GNNs.

**Theorem B.1** *Given an arbitrary non-attributed graph with $n$ nodes, if the degree of each node in the graph is between $1$ and $\mathcal{O}(\log^{\frac{1-\epsilon}{2k}}(n))$, for any constant $\epsilon > 0$, given $m$ virtual nodes which evenly divide the node set into $m$ clusters, there are $\omega\left((m-1)^2(\frac{n^\epsilon}{m}-1)^3\right)$ many pairs of non-isomorphic links $(u, w), (v, w)$ such that a $k$-layer 1-WL-GNN gives $u, v$ the same representation, while using $m$ virtual nodes give $(u, w), (v, w)$ different representations.*

The theorem says that 1-WL-GNN with virtual nodes can discriminate many links that 1-WL-GNN cannot discriminate. On the other hand, it is intuitive adding virtual nodes can be at least as powerful as 1-WL-GNNs since it keeps all other components. If there are links that 1-WL-GNN can discriminate we only need to assign the related nodes to the same virtual nodes, so that the virtual-nodes-based method can also discriminate them.

### B.1.1  Proof of Theorem B.1

The proof can be separated into two steps. The first step is to prove that there exist $n/o(n^{1-\epsilon}) = \omega(n^\epsilon)$ many nodes that are locally $h$-isomorphic (which means their $h$-hop enclosing subgraphs are isomorphic). This step is same as the proof of Theorem 2 in [6], so we omit the details here. The basic idea is to expand the $h$-hop enclosing subgraph $G^h(v)$ of $v$ to another subgraph $\tilde{G}^h(v)$ and then use the pigeon hole principle to count the possible isomorphic $\tilde{G}^h(v)$. After getting these locally isomorphic nodes, we denote the set of these nodes as $V_{iso}$. The second step is to find the non-isomorphic links.

**Step 2.** We partition $V_{iso} = \cup_{i=1} V_i$ where $V_i$ is the subset of nodes connected to virtual node $s_i$. To be simple, we call each $V_i$ a cluster, and the sizes of different clusters are assumed to be the same $|V_i| = |V_{iso}|/m$. Consider two nodes $u \in V_i$ and $v \in V_j$ from different clusters. Since both of them are in $V_{iso}$, they have identical $h$-hop neighborhood structures, and $h$-layer 1-WL-GNN will give them the same representations. Then let us select another node $w$ in $V_i$, $h$-layer 1-WL-GNN will also make $(u, w)$ and $(v, w)$ have the same representation.

However, if we use virtual nodes to label nodes and give them additional features, because $u, w$ are in the same cluster while $v, w$ belong to different clusters, $(u, w)$ will have different representation from $(v, w)$. Now we count the number of such non-isomorphic link pairs $Y$, we obtain:

$$Y \geq \prod_{i,j=1,j\neq i}^{m} |V_i||V_i - 1||V_j|$$
$$= \frac{1}{2}m(m-1)\left(\left(\frac{|V_{iso}|}{m} - 1\right)\left(\frac{|V_{iso}|}{m}\right)^2\right)$$

Taking $|V_{iso}| = \omega(n^\epsilon)$ into the above in-equation, we get

$$Y \geq \frac{1}{2}m(m-1)\omega\left((\frac{n^\epsilon}{m} - 1)^3\right)$$
$$= \omega\left((m-1)^2(\frac{n^\epsilon}{m} - 1)^3\right)$$

## B.2 Node Influence

In the following, without loss of generality, we take a $k$-layer GCN [30] as the example, and hence consider layers described as follows:

$$h_v^l = \text{ReLU}(W_l \frac{1}{deg(v)} \sum_{u \in N(v)} h_u^{l-1}).$$

**Influence Score .** We measure the sensitivity of a node $y$ on a node $x$ by the *influence score* [8] $I(x,y) = e^{\text{T}} \frac{\partial h_x^k}{\partial h_y^0}$; $e$ is a vector of all ones and $h_x^k$ is the embedding of $x$ at the $k^{th}$ layer. The *influence score is known to be proportional in expectation to the $k$-step random walk distribution $P_{rw}$ from $x$ to $y$:* [4]

$$\mathbb{E}[I(x,y)] \propto P_{rw}(x \to y, k) = \sum_{r \in R^k} \prod_{\ell=1}^{k} \frac{1}{deg(v_r^\ell)}, \tag{3}$$

$(v_r^0, v_r^1, ..., v_r^k)$ are the nodes in the path $r$ from $x := v_r^0$ to $y := v_r^k$, $R^k$ is the set of paths of length $k$. In what follows, we exploit the relationship between the influence score and the probability $P_{rw}$ and argument in terms of the latter. In particular, we will show how $P_{rw}$ changes in view of virtual nodes. Note that we assume all the paths of message passing have the same probability. We assume a self-loop at each regular graph node, this is standard and supported by Equation (1). Hence, the denominator in the above equation changes slightly:

$$P_{rw}(x \to y, k) = \sum_{r \in R^k} \prod_{\ell=1}^{k} \frac{1}{deg(v_r^\ell) + 1}. \tag{4}$$

We neglect the self-loops with virtual nodes only for reasons of readability. But it can be readily checked that the later equations hold similarly with an additional "+1" in denominators. For simplicity, we further consider the graph to be $r$-regular; in the standard case without virtual nodes, Equation (4) then simplifies to:

$$P_{rw}(x \to y, k) = \frac{|R^k|}{(r+1)^k}. \tag{5}$$

We hypothesize that we can come to similar conclusions in a general graph with average degree $r$.

**Impact of One Virtual Node.** We focus on the message passing between two nodes $x$ and $y$, in layer $k$ and calculate $P_{rw}^s(x \to y, k)$, the influence score in the setting with virtual nodes, here with one, $s$. In particular we assume $x, y \in V$ and hence $x, y \notin \{s\}$. We argument inductively, based on $k$ and, for each GNN layer, separate the impact of the messages coming from the virtual node. For $k = 1$, all the nodes can be classified into two cases: if $y$ is not connected to $x$, the influence changes from 0 to $\frac{1}{(r+2)}$; if $y$ is connected to $x$, the influence score is:

$$P_{rw}^s(x \to y, 1) = P_{rw}(x \to y, 1) + P_{rw}(x \to s, s \to y)$$
$$= \frac{1}{(r+2)} + \frac{1}{(r+2)|V|}. \tag{6}$$

Note that the probability for $x \to s$ is the same as from $x$ to any other neighbor, $\frac{1}{|V|}$ for $s \to y$ follows from the $|V|$ connected nodes at $s$.

For $k \geq 2$, we obtain:

$$P_{rw}^s(x \to y, k) = P_{rw}(x \to y, k) + P_{rw}(x \to s, s \to y) P_{rw}^s(x \to y, k-1)$$
$$= \frac{|R^k|}{(r+2)^k} + \frac{1}{(r+2)|V|} P_{rw}^s(x \to y, k-1). \tag{7}$$

---

[4] See Theorem 1 in [8]. Note that the theorem makes some simplifying assumptions that all paths in the computation graph of the model are activated with the same probability of success. Nevertheless, empirical experiments presented in [8] confirm that the theory is close to what happens in practice. In addition, the GCN is assumed to use a simple average as AGG function. However, the factor in the equation can be easily adapted to other GNNs.

**Multiple Virtual Nodes.** In view of multiple virtual nodes, the above analysis gets more appealing.

We continue along the above lines and assume there is a shortest path of length $\leq k$ between $x$ and $y$. If $x$ and $y$ connect to the same virtual node $s$, then Equation (7) changes as follows:

$$P_{rw}^{ms}(x \to y, k) = \frac{|R^k|}{(r+2)^k} + \frac{1}{(r+2)|C_s|} P_{rw}^{ms}(x \to y, k-1). \tag{8}$$

Since the set $C_s$ of nodes connecting to $s$ is much smaller than $V$, i.e., $|C_s| << |V|$, *the impact of multiple virtual nodes on the influence score is greater than that of a single virtual node.* In case that $x$ and $y$ do not connect to the same virtual node, the probability just slightly decreases. The maximum possible decrease occurs when no nodes in the path between $x$ and $y$ are connected to a common virtual node, including $x$ and $y$: $\delta_{wc} = \frac{|R^k|}{(r+1)^k} - \frac{|R^k|}{(r+2)^k}$; here we subtract from the regular $P_{rw}$ our $P_{rw}^{ms}$, in which the second (virtual node) component is 0.

**Impact on Over-Smoothing** The idea is to show that multiple virtual nodes help to preserve local information at the graph nodes. To this end, we consider the influence of $x$ onto itself. For the setting with a single virtual node and $k = 1$, the change in influence score is

$$
\begin{aligned}
\delta^s(x, k = 1) &= P_{rw}^s(x \to x, 1) - P_{rw}(x \to x, 1) \\
&= \frac{1}{(r+2)} + \frac{1}{(r+2)|V|} - \frac{1}{(r+1)} \\
&= \frac{(1 + \frac{1}{|V|})(r+1) - (r+2)}{(r+1)(r+2)} \\
&= \frac{\frac{1}{|V|}(r+1) - 1}{(r+1)(r+2)} < 0.
\end{aligned}
\tag{9}
$$

This means the node will preserve less information for itself at layer considering the message coming from the single virtual node. However, in view of multiple virtual nodes, we come to a different conclusion.

$$
\begin{aligned}
\delta^{ms}(x, k = 1) &= \frac{1}{(r+2)} + \frac{1}{(r+2)|V|} - \frac{1}{(r+1)} \\
&= \frac{(1 + \frac{1}{|C_s|})(r+1) - (r+2)}{(r+1)(r+2)} \\
&= \frac{\frac{1}{|C_s|}(r+1) - 1}{(r+1)(r+2)}
\end{aligned}
$$

Since $|C_s| << |V|$, we obtain $\delta^{ms}(x, k = 1) >> \frac{\frac{1}{|V|}(r+1) - 1}{(r+1)(r+2)}$, which means *we can preserve much more local information than in the setting with a single virtual node.* Especially, when $|C_s| \leq r + 1$, the self-transition probability is even higher than in the original setting without virtual nodes. We encounter this scenario practically especially with dense graphs. This fits nicely since these graphs are particularly prone to over-smoothing and, as shown in [8, 16], *additional capability to preserve local information in message passing steps helps to reduce over-smoothing.*

For $k \geq 2$,

$$\delta^{ms}(x, k) = \frac{|R^k|}{(r+2)^k} + \frac{1}{(r+2)|C_s|} P_{rw}^{ms}(x \to x, k-1) - \frac{|R^k|}{(r+1)^k}$$

Assume $P_{rw}^{ms}(x \to x, k-1) > P_{rw}^s(x \to x, k-1)$, since $|C_s| < |V|$, then we get

$$\delta^{ms}(x, k) < \frac{|R^k|}{(r+2)^k} + \frac{1}{(r+2)|V|} P_{rw}^s(x \to x, k-1) - \frac{|R^k|}{(r+1)^k} = \delta^s(x, k).$$

Adding the condition that $\delta^{ms}(x, k = 1) > \delta^s(x, k = 1)$, we know that for any $k$, multiple virtual nodes can preserve more local information than single nodes.

**Table 4:** Data. All graphs are undirected, have no edge features, and all but `ddi` have node features.

|  | #Nodes | #Edges | Average Node Deg. | Average Clust. Coeff. | MaxSCC Ratio | Graph Diameter |
|---|---|---|---|---|---|---|
| `ddi` | 4,267 | 1,334,889 | 500.5 | 0.514 | 1.000 | 5 |
| `collab` | 235,868 | 1,285,465 | 8.2 | 0.729 | 0.987 | 23 |

### B.3   Relationship with Labeling Tricks

Although the concept of link representation is from [6], we would like to clarify that *our labeling strategy is not a valid labeling trick* by the definition of [6].

Consider an undirected graph $G$ as described in Section 2. In addition, the tensor $\mathbf{A} \in \mathbb{R}^{n \times n \times k}$ contains all node and edge features (if available). The diagonal components $\mathbf{A}_{v,v,:}$ denote the node features, while the off-diagonal components $\mathbf{A}_{u,v,:}$ denote the edge features of edge $(u,v)$. The labeling trick uses a target node set $S \subseteq V$ and a labeling function to label all nodes in the node set $V$ and stack the labels with $\mathbf{A}$. A valid labeling trick must meet two conditions: (1) the nodes in $S$ have different labels from the rest of the nodes, (2) the labeling function must be permutation invariant.

Using virtual nodes is not a valid labeling trick in the following two aspects: First, the virtual node set $S$ is not a subset of graph nodes $V$, and we use addition instead of concatenation. Even if we extend $V$ to $V \cup S$, our labeling strategy still does not fit the permutation-invariant requirement. Nevertheless, it can achieve similar effects in learning structural link representations.

## C   Additional Experimental Details and Results

### C.1   Data Statistics

Data Statistics of the `ddi` and `collab` are shown in Table 4. From the table, we can see both of the datasets have good clustering structure. `ddi` is extremely dense and `collab` is sparser. That is interesting to note that on the denser `ddi`, our virtual nodes approaches achieved better performance gain.

### C.2   Model Configurations and Training

We trained all models for 80 runs using the Bayesian optimization provided by wandb[5] to select hyperparameters from the following pool.

| | |
|---|---|
| hidden dimension | 32, 64, 128, 256 |
| learning rate | 0.1, 0.05, 0.01, 0.005, 0.001, 0.0005, 0.0001 |
| dropout | 0, 0.3, 0.6 |
| # of layers | 1-7 |
| # of virtual nodes (random) | 1-10 |
| # of virtual nodes | 1,2,4,8,16,32,64 |
| SGC - K | 2-7 |
| APPNP - $\alpha$ | 0.05, 0.1, 0.2, 0.3 |
| GNN-GDC - k | 64, 128 |
| GNN-GDC - $\alpha$ | 0.05, 0.1, 0.2, 0.3 |

Please note that we considered the wide ranges of values only in order to find a good general setting. For practical usage a hidden dimension of 256, learning rate of 0.0001, and dropout of 0.3 should work well; only on the small graphs a dropout of 0 might work better. As usual, the number of layers depends on the type of data; however, note that the virtual nodes make it possible to use more that then the usual 2-3 layers. Generally, higher numbers of virtual nodes work better, in line with our theoretical results.

Also note that we used less virtual nodes in the selection for the models -RM since preliminary results showed that larger numbers did not change the results greatly – probably due to the randomness. We

---

[5]https://wandb.ai/site

**Figure 3:** Performance depending on layers: Hits@k and time per epoch (sec.); `ddi` (left), `collab`.

used maximally 64 virtual nodes due to memory issues with larger numbers (e.g., 128), especially on the larger datasets. We used 500 epochs with a patience of 30. Furthermore, for `collab`, we used the validation edges during testing (OGB contains both settings, with and without them).

We tuned all models for 80 runs, and thereafter ran the models with the best 3 configurations for 3 runs and chose the best of these model as the final model (configuration). We trained as suggested by the OGB (e.g., the splits, negative sampling) but used a batch size of $2^{12}$.

## C.3  Additional Results

**Impact of Virtual Nodes on Number of GNN Layers and Efficiency, Figure 3.** For the virtual nodes models, the scores increase with the number of layers for a longer time, GCN drops earlier. On `ddi`, GCN-VN and -CM reach their best scores at 6 and 8 layers, respectively, which is remarkable for that very dense dataset, being prone to over-smoothing. On `collab` it is the other way around. The figure also gives an idea about the runtime increase with using virtual nodes. It compares the 6-layer models, and shows the 4-layer GCN-CM which obtains performance similar to the 6-layer GCN-VN.

