# OpenReview forum: "An Analysis of Virtual Nodes in Graph Neural Networks for Link Prediction (Extended Abstract)"
_logconference.io/LOG/2022/Conference — LoG 2022 Oral_

### Official Review · Reviewer_TN6Y · 2022-10-17

**Overall Score:** 6
**Confidence:** 4

**Review:**

Summary: The paper studies the problem of link prediction by graph neural networks (GNNs) by adding multiple virtual nodes to the graph. Adding virtual nodes can help in addressing some existing limitations of GNNs (e.g., over-smoothing, lack of expressiveness) and has been shown effective for graph classification.  The proposed method shows that virtual nodes might be helpful for communicating between distant nodes in the same cluster with significant theoretical analyses. The paper also provides rigorous empirical analyses using the Open Graph Benchmark datasets.

Comments:

Overall, I enjoyed reading this paper. I am inclined towards acceptance.  I will list down the positives and some comments that might help further improve the paper.

Positives:
1. The paper is well-written and easy to follow. Also, the problem is clearly defined, and the method is supported by enough intuitions.

2. The theoretical analyses of the proposed method of adding virtual nodes in the context of the link prediction problem are novel.

3. The experiments show that the method improves the performance of the base GNNs. It also includes an extensive ablation study and exhaustive datasets.

Comments/Questions:
1. The paper compares the baselines from the leaderboard from the OGB. However, it does not really show the SOTA method from OGB. Please refer to [1].
2. Figure 1 is a good example of how virtual nodes should work. What happens if the nodes use positional embeddings? Do we still need virtual nodes? It would be useful to have a discussion on it.
3. The experiments are showing improvements over the basic GNN models. While these show the importance of the method,how will the method improve the more complex/sophisticated methods such as in [1]?


[1] Wang, Zhitao, Yong Zhou, Litao Hong, Yuanhang Zou, and Hanjing Su. "Pairwise Learning for Neural Link Prediction." arXiv preprint arXiv:2112.02936 (2021).

---

### Official Review · Reviewer_dCPX · 2022-10-17

**Overall Score:** 8
**Confidence:** 4

**Review:**

This paper analyzes the effect of virtual nodes in graph neural networks for link prediction. Specifically, the authors generate virtual nodes for each group of nodes based on the graph clustering algorithm and then connect nodes to the virtual nodes to utilize it in GNNs. They also provide the first theoretical analysis of the benefits of virtual nodes regarding the expressiveness of the learned link presentation and the potential impact on under-reaching and over-smoothing. Experiments on two datasets show the effectiveness and guidelines of using virtual nodes in GNNs for link prediction.

Strengths
- The analysis of using virtual nodes in GNNs for link prediction is novel.
- The authors first theoretically analyze the effectiveness of virtual nodes in terms of the expressiveness of link representation.
- The experimental results show that employing virtual nodes in GNNs can be a meaningful direction for link prediction.

Weakness
- I understand that it is an extended abstract, but it would’ve been better if there was an analysis of more datasets.

---

### Official Review · Reviewer_fwHQ · 2022-10-18

**Overall Score:** 8
**Confidence:** 3

**Review:**

**Summary**

The authors investigate the effectiveness of creating (multiple) virtual nodes in GNN for link prediction tasks and provide theoretical analysis and comprehensive empirical study, including various variants and hyper-parameter searches.

**Strength**

- The paper is well motivated and written, with many interesting and honest details, which are very much appreciated.
- While intuitively, virtual nodes will increase the impact of node influence, it is helpful to see a formal and theoretical analysis.
- By the same token, even though we know virtual nodes can increase the GNN expressiveness, it's useful to see the empirical evidence, especially for link prediction tasks, which might be less studied.
- The paper benchmarked many GNNs and their variants WITH hyper-parameter searches.

**Weakness**

It's relatively clear that virtual nodes help with link predictions, but there are many mixed results and unclear trends in the analysis, which might be addressable from the following axis:
- More benchmark datasets. I understand the authors choose the smaller-scale dataset on OGB due to memory constraints, but it might be useful to look at subsets of other larger-scale datasets to get a clearer answer?
- Sufficient hyper-parameter search. It's unclear to me whether 80 runs per model are enough for hyper-parameter search, especially since the noise in the performance trend can be the result of insufficient search budgets. A plot with best performance vs. # experience ran would be helpful to address this concern.

**Recommendation**
For the above reasons, I am giving a weak acceptance score for the paper. The analysis would be interesting and useful for the community, but additional benchmark datasets and (proof of) sufficient hyperparameter search might be necessary.

I will consider raising my scores if the authors can:
1) incorporate two more datasets
2) address my concerns around insufficient hyperparameter search

**Nit Picks**

The introduction of "virtual nodes" via clustering algorithm is not exactly new and has been explored in "Hierarchical Graph Representation Learning with Differentiable Pooling" by Ying et al. (2018).

---

### Official Review · Reviewer_vWdw · 2022-10-20

**Overall Score:** 8
**Confidence:** 3

**Review:**

The authors proposed to add virtual nodes for link prediction tasks and analyzed the impact of these additional virtual nodes based on a clustering method. The authors evaluated the proposed methods on two large-scale graphs and showed improvement.

The experiments and analysis are sufficient for an extended abstract. But I still have several questions:
1. How's the complexity of the clustering algorithm? Are they efficient enough compared with the training time?
2. How's the clustering algorithm affect the performance? Or does only the number of virtual nodes affect the results?
3. I wonder why the training used Bayesian optimization instead of gradient-based methods.

---

### Meta-Review · Area_Chair_dwJk · 2022-11-08

**Confidence:** 4
**Recommendation:** Accept for spotlight

**Meta Review:**

Good paper with enthusiastic reviews from all reviewers. Authors are encouraged to incorporate how to best use virtual nodes in "node prediction" tasks as well, to make the impact of this paper much larger.

---

### Decision · Program_Chairs · 2022-11-22

Accept (Oral)